# Factors associated with antenatal exercise in Arba Minch town, Southern Ethiopia: A community-based cross-sectional study

**Maechel Maile Beyene**[1]*, **Mulugeta Shegaze Shimbre**[2], **Gebresilasea Gendisha Ukke**[1], **Mathewos Alemu Gebremichael**[2], **Mekdes Kondale Gurara**[2]

1 Department of Midwifery, College of Medicine and Health Sciences, Arba Minch University, Arba Minch, Ethiopia, 2 Department of Public Health, College of Medicine and Health Sciences, Arba Minch University, Arba Minch, Ethiopia

* maechelmaile11@gmail.com

**Data Availability Statement:** All relevant data are available on the paper and the anonymized SPSS data set of study participants was uploaded as a supporting information file.

## Abstract

### Background

Many health risks in pregnant women and their foetuses can be reduced by practicing antenatal exercise. However, the adequate practice of antenatal exercise among pregnant women is low in Ethiopia. Therefore, this study aimed to assess the practice of antenatal exercise and its associated factors among pregnant women in Arba Minch town.

### Methods

A community-based cross-sectional study design was conducted. Data were collected by using a structured questionnaire from 422 pregnant women selected by a simple random sampling technique. Descriptive statistics were computed and a binary logistic regression model was fitted. In multivariable logistic-regression adjusted odds ratio (AOR) with 95% confidence intervals were used to determine the strength of associations. The significance level was declared at a p-value < 0.05.

### Results

Among 410 participants, 32.9% (95% CI 28%-37%) adequately practiced antenatal exercise. Factors negatively associated with an adequate antenatal exercise were husband's primary school level [Adjusted odds ratio (AOR) = 0.3, (95% CI: 0.1, 0.7)], history of miscarriage [AOR = 0.3, (95% CI: 0.1, 0.7)], inadequate knowledge [AOR = 0.2, (95% CI: 0.1, 0.3)], and unfavorable attitude [AOR = 0.3, (95% CI 0.2, 0.5)]. Whereas, factors positively associated with an adequate antenatal exercise were employment status of women [AOR = 4.8, (95% CI: 1.8, 13.1)], and a practice of regular exercise before current pregnancy [AOR = 1.9, (95% CI: 1.1, 3.2)].

### Conclusions

The findings of this study indicated that adequate practice of antenatal exercise was found to be low. Appropriate measures should be taken to improve the husband's educational

**Funding:** Funding was obtained from Arba Minch University, college of Medicine and Health sciences.https://www.amu.edu.et/ The funders had no role in study design, data collection and analysis, decision to publish, or preparation of the manuscript.

**Competing interests:** The authors have declared that no competing interests exist.

**Abbreviations:** ACOG, American Congruence of Obstetrics and Gynecology; ANC, Antenatal Care; ANEx, Antenatal Exercise; CI, Confidence Interval; KAP, Knowledge Attitude and Practice; OR, Odd Ratio; SNNPR, Southern Nation's Nationalities Peoples Region; SPSS, Statistical Package for Social Sciences; WHO, World Health Organization.

level, mother's occupation, knowledge, and attitudes towards antenatal exercise. Special consideration should be given to those with a history of miscarriage and women should be encouraged to practice regular exercise before pregnancy.

## Introduction

Physical activity is a bodily movement produced by the contraction of skeletal muscles in all stages of life while exercise is a structured, planned, and repetitive movement produced by skeletal muscles [1]. Pregnant women are encouraged to continue and maintain moderate-intensity physical exercise using both aerobic and muscle conditioning activities during pregnancy in the absence of medical or obstetrical complications under the guidance of health care providers [2–4].

Regular participation in exercise has become an important component of a healthy lifestyle, so antenatal exercise has become a fundamental aspect of women's lives and an important constituent of antenatal care [5,6]. American College of Obstetrics and Gynecology (ACOG) recommended that low to a moderate regular impact exercise regime for pregnant mothers performed for at least 20 to 30 minutes on most days of the week and gradually progressed over a period of time can be followed to improve overall fitness [4]. These exercises include aerobic exercises such as aerobics, swimming, cycling, walking, dancing, core stability, pelvic floor exercises, breathing exercises, postural education, back care foot & leg exercises, and pelvic tilting [4].

Scientific pieces of the literature showed that, in most cases, appropriate antenatal exercise is a safe and effective way to gain many physical and mental health benefits such as avoiding sedentary- and obesity-associated risks for both the mother and foetus [7]. Numerous health benefits of antenatal exercise during pregnancy were documented such as the reduced risk of excessive gestational weight gain, gestational diabetes, preeclampsia, preterm birth, varicose veins, deep vein thrombosis, reduced length of labour, fatigue, stress, anxiety, and depression, as well as decrease delivery complications and improved well-being [8].

Furthermore, antenatal exercise improves the functioning of the foetal and neonatal cardiac autonomic nervous system, normalizes birth weights, and reduces adiposity at birth and in early childhood. Additionally, babies born seem to be calmer, are leaner, more intelligent with improved neurological and mental development and their children had significantly higher scores on oral language and general intelligence tests [9].

Physical inactivity in the general population and lack of antenatal exercise was taken as the leading risk factors for death estimating 3.2 million deaths worldwide and the fourth leading risk factor for early mortality worldwide [10]. It was reported that lack of antenatal exercise increased the prevalence of chronic diseases such as cardiovascular disease, type 2 diabetes, osteoporosis, cancer, and their risk factors such as raised blood pressure, postpartum haemorrhage, raised blood sugar, and overweight [11].

Nonetheless, scholars around the world documented that antenatal exercise is not adequately practiced and does not meet sufficient exercise recommendations [12–16]. In Indian, 35.2% [12], in Campinas Brazil, 29% [13], in Colombo Sri Lanka, 13.6% [14], in Pakistan, 20% [15], 27.2% in Serbia [17], and in Gondar town, Northern Ethiopia, 37.9% [16] of the study participants practiced antenatal exercise. The participation of pregnant women in physical activity during pregnancy in Africa is also low [18]. To guide contextual interventions, examining the physical activity during pregnancy is recommended [18]. However, the status of antenatal exercise practice in Southern Ethiopia has not been assessed yet. Hence, this study

was aimed to assess the adequate practice of antenatal exercise and associated factors among pregnant women in Arba Minch town, Southern Ethiopia.

## Methods and materials

### Study design, period, setting, and population

The community-based cross-sectional study design was conducted from December 1st to 30th, 2019 GC in Arba Minch town, Southern Ethiopia. Arba Minch town is located 454 Km to South of Addis Ababa (capital city of Ethiopia) and about 280 Km from the regional town of Hawassa. According to the 2007 census, the total population of the town was 74,879, of whom 39,208 were men and 35,671 were women [19]. A total of 23.5% were found to be reproductive age group, among these, 4,084 women were pregnant. The numbers of health institutions in Arba Minch town were 1 governmental general hospital, 3 health centers, 11 health posts, 33 private clinics, and 13 drug stores.

The source population was all pregnant women in Arba Minch town. All pregnant women who lived in Arba Minch town at least for six months were included while pregnant women who had medical or obstetric complications and serious psychological conditions were excluded.

### Sample size determination and sampling technique

The sample size was determined by using single population proportion formula through Open Epi, Version 3, open-source considering the following assumptions: the adequate practice of antenatal exercise to be 50%, 95% confidence level of certainty (alpha = 0.05), 5% marginal error, and adding 10% non-response rate. The final sample size was determined as 422. There were four sub-cities in the town which were divided into 11 *Kebeles (small administrative unit of Ethiopia)*. The sample size was allocated to all *Kebeles* proportionally to their number of pregnant women size. Each pregnant woman was selected by a simple random sampling technique using a family folder as a sampling frame (Fig 1).

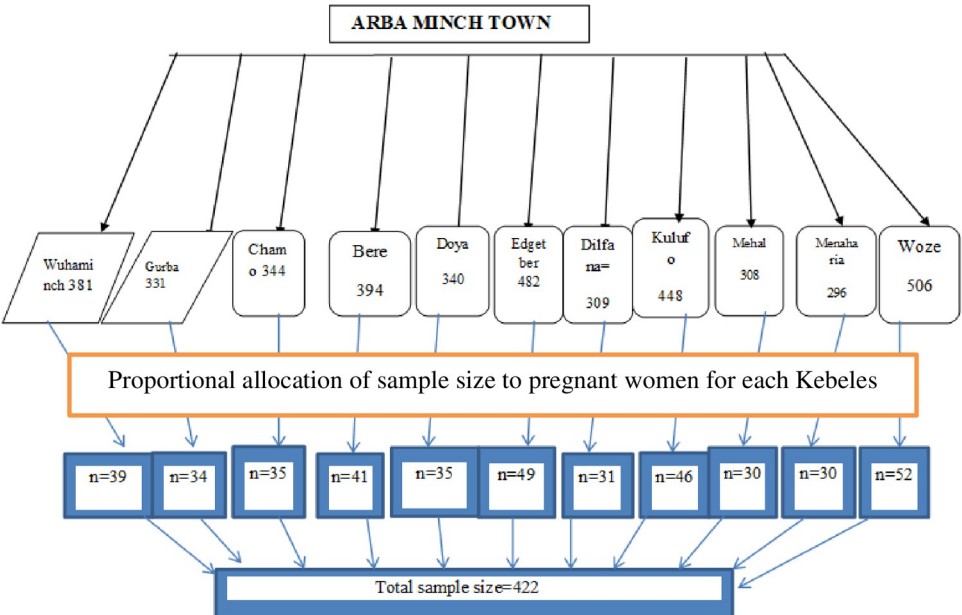

**Fig 1. Schematic representation of sampling procedure of pregnant women in Arba Minch town, Southern Ethiopia 2020.**

The institutional review board (IRB) of Arba Minch University, College of Medicine and Health sciences ethically approved the study with a letter reference number (IRB/129/12 on the date of 14/11/2019). Permissions were obtained from each Kebele. Before enrolment, the pregnant women were informed about the objectives of the study, and its importance and informed verbal consent was taken from all study participants before data collection because some of the study participants cannot read and write and their consent was recorded in the questionnaire. For minors, the consent was taken from parents.

## Study variables

The dependent variable of this study was the practice of antenatal exercise. Independent variables were: socio-demographic characteristics like age, occupation, religion, monthly income, educational level, employment and marital status, obstetric characteristics like gravidity, gestational ages, parity, and the number of children and history of miscarriage, the practice of exercise before pregnancy and knowledge and attitude towards practices of antenatal exercise.

## Data collection tools and procedures

Data were collected using a pre-tested structured questionnaire by face-to-face interview. The questionnaire was designed in English from kinds of literature on related topics [12,14,20] and based on ACOG recommendations of antenatal exercises [4] and was translated to Amharic version for better understanding by data collectors and interviewees. A total of eight data collectors (4 diploma nurses and 4 diploma midwives) were involved in data collection and two master holders in public health were assigned for supervision. The questionnaire contained socio-demographic characteristics, obstetric characteristics, the practice of antenatal exercise, and knowledge and attitude towards practices of antenatal exercise.

Those participants who exercised at least three times a week for a minimum of 20 minutes according to ACOG recommendations [4] were categorized to have an adequate practice of antenatal exercise. Knowledge of antenatal exercise was measured by asking sixteen questions about the benefit and contraindication of antenatal exercise with categorical responses of "yes" and "no" with an item score of "1", and "0" respectively. Then, the sum was computed and those who responded correctly to the mean and above value were considered as they had adequate knowledge and those who scored less than the mean value were labelled as they had inadequate knowledge about antenatal exercise. The attitude of the woman concerning performing antenatal exercise during pregnancy was measured by asking 8 questions and classified as favourable or unfavourable. Those who answered to 8 attitude questions and scored greater than and equal to mean values were categorized to have favourable attitudes whereas, those who scored less than mean values were categorized to have unfavourable attitudes.

## Data quality control

Before actual data collection time, a pre-test was done on 5% [21] women in Mirab Abaya town using the Amharic questions and modifications were done. The training was given to data collectors and supervisors before the beginning of actual data collection. Supervisors closely followed the data collection process. The completed questioners were checked for completeness and consistency on daily basis.

## Data analysis

The collected data were coded and entered into Epi Data version 4.6 then exported to SPSS version 25 for analysis. Descriptive statistics like frequencies, percentages, means, and standard

deviations were computed. The binary logistic regression model was used. Variables with a p-value ≤ 0.25 in the bi-variable analysis were entered into the multivariable analysis. In addition, the context and findings of previous studies were considered in the identification of candidate variables for multi-variable logistic-regressions to adjust for possible confounding variables. AOR with 95% CI was used to measure the strength of associations and a p-value of < 0.05 was used to determine the presence of association with the outcome variable. Hosmer and Lemeshow model fitness test was used to assess model fitness and it was a good fit.

# Results

## Socio-demographic characteristics of study participants

Of the 422 women interviewed, 410 (97.2%) provided complete information and were used in these analyses. The mean ± SD (standard deviation) age of the participants was 25.8 ± 5.2 years and the minimum and maximum ages were 16, and 42 years, respectively. The majority of women were within the age category of ≥ 25 years (56.8%). More than half of the study participants 231 (56.3%,) were protestant Christians. A majority of the participants 390 (95.1%) were married. Most women 176 (42.9%) attended primary school while around 30% of their husbands attended their education in college and above. Concerning occupation, 335 (81.7%) of the women were none employed. The monthly mean income of the study participants was 1,347 Ethiopian birr. More than half (64.1%) of the respondents belong to the income level of ≥ 1,347 Ethiopian birrs (Table 1).

## Obstetric characteristics of the respondents

The majority of the participants were multigravida 272 (66.3%). Nearly half (46.6%) of them had 1–2 number of living children while more than 90% of the women had no history of miscarriage. Forty-six percent of the women were within four to six months of pregnancy followed by seven to nine months of pregnancy. Half of the pregnant women commenced their antenatal care and visited the health institution 1–2 times (Table 2).

## Sources of information for antenatal exercise

A total of 29.3% of respondents revealed that family/friends were the commonest sources of information (Table 3).

## Pregnant women's knowledge about antenatal exercise

From total respondents, 233(56.8%) of the pregnant women reported that antenatal exercise reduces the risk of back pain. The majority, 219(53.4%) of pregnant women responded that excessive weight can be prevented by antenatal exercise. About 235(57.3%) of pregnant women reported that antenatal exercise increases energy and stamina. A total of 217(52.9%) women indicated that antenatal exercise helps to cope with labour and delivery pain. Regarding the contraindication of antenatal exercise, a total of 199 (48.53%) pregnant women reported that vaginal bleeding as a contraindication for antenatal exercise. The mean score value of pregnant women's knowledge about antenatal exercise was 7.99 out of 16. Among all participants, only 190 (46.34%) scored above the mean value and had adequate knowledge about antenatal exercise (Table 4).

## Attitude towards antenatal exercise among pregnant women

Among total respondents, 17(17.8%) of pregnant women strongly agree that antenatal exercise during pregnancy is necessary. From total respondents, only 31(7.6%) strongly disagree that

**Table 1. Socio-demographic characteristics of pregnant women in Arba Minch town, Southern Ethiopia, 2020 (n = 410).**

| Variables | Categories | Frequency (n) | Percentage (%) |
|---|---|---|---|
| Age (in years) | < 25 | 177 | 43.2 |
| | ≥ 25 | 233 | 56.8 |
| Religion | Orthodox | 137 | 33.4 |
| | Protestant | 231 | 56.3 |
| | Catholic | 10 | 2.4 |
| | Muslim | 32 | 7.8 |
| Marital status | Married | 390 | 95.1 |
| | Divorced | 13 | 3.2 |
| | Widowed | 7 | 1.7 |
| Woman educational status | Unable to read and write | 36 | 8.8 |
| | Able to read and write | 24 | 5.9 |
| | Primary school | 176 | 42.9 |
| | Secondary school | 88 | 21.5 |
| | College and above | 86 | 20.9 |
| Husband educational status | Unable to read and write | 13 | 3.2 |
| | Able to read and write | 52 | 12.7 |
| | Primary school | 107 | 26.1 |
| | Secondary school | 115 | 28.0 |
| | College and above | 123 | 30.0 |
| Occupation | Employed | 75 | 18.3 |
| | Non employed | 335 | 81.7 |
| Income (ETB) | < 1,347 | 147 | 35.9 |
| | > = 1,347 | 263 | 64.1 |

ETB-Ethiopian Birr (1ETB = 0.0241243 USD).

**Table 2. Obstetrical characteristics of the study participants in Arba Minch town, Southern Ethiopia, 2020 (n = 410).**

| Characteristics | Categories | Frequency (n) | Percent (%) |
|---|---|---|---|
| Gravidity | Prim gravida | 138 | 33.7 |
| | Multigravida | 272 | 66.3 |
| Parity | Nulliparous | 136 | 33.2 |
| | Prim-parous | 135 | 32.9 |
| | Multiparous | 140 | 34.1 |
| Number of alive child they have | No child | 135 | 32.9 |
| | 1–2 child | 191 | 46.6 |
| | >2 children | 83 | 20.2 |
| History of miscarriage | Yes | 39 | 9.5 |
| | No | 371 | 90.5 |
| Gestational age | <4 months | 34 | 8.3 |
| | 4–6 months | 189 | 46.1 |
| | 7–9 months | 187 | 45.6 |
| ANC follow up | Not started | 102 | 24.9 |
| | 1–2 times | 208 | 50.7 |
| | Three and above | 100 | 24.4 |

ANC-Antenatal care.

**Table 3. Sources of information about antenatal exercise among pregnant women in Arba Minch town Southern, Ethiopia, 2020.**

| Source | Frequency | Percent (%) |
|---|---|---|
| Health care provider | 97 | 23.7% |
| Health extension workers | 99 | 24.1% |
| Family/friends | 120 | 29.3% |
| Mass media (Tv, radio) | 43 | 10.5% |
| Internet | 38 | 9.3% |

**Table 4. Knowledge of antenatal exercise of study participants in Arba Minch town, southern, Ethiopia, 2020 (n = 410).**

| Benefits and contraindication of antenatal exercise | Response | Frequency | Percent |
|---|---|---|---|
| Reduces risk of back pain | Yes | 233 | 56.8 |
| | No | 197 | 43.7 |
| Prevents excessive weight gain | Yes | 219 | 53.4 |
| | No | 191 | 46.6 |
| Increases energy and stamina | Yes | 235 | 57.3 |
| | No | 175 | 42.7 |
| Help cope with labour and delivery pain | Yes | 217 | 52.9 |
| | No | 193 | 47.1 |
| Can reduces risk of diabetes mellitus | Yes | 143 | 34.9 |
| | No | 267 | 65.1 |
| Can decrease high blood pressure during pregnancy | Yes | 196 | 47.8 |
| | No | 214 | 52.2 |
| Helps more rapid postnatal recovery | Yes | 190 | 46.3 |
| | No | 220 | 53.7 |
| Can prevents antenatal and postnatal depression | Yes | 213 | 52.0 |
| | No | 197 | 48 |
| Benefits general health and development of the baby | Yes | 241 | 58.8 |
| | No | 169 | 42 |
| Contraindications | | | |
| Vaginal bleeding | Yes | 199 | 48.5 |
| | No | 211 | 51.5 |
| Uterine contractions | Yes | 185 | 45.1 |
| | No | 225 | 55.9 |
| Chest pain | Yes | 196 | 47.8 |
| | No | 214 | 52.2 |
| Difficulty of breathing | Yes | 205 | 50 |
| | No | 205 | 50 |
| Premature labour | Yes | 201 | 49.0 |
| | No | 209 | 51 |
| Poorly controlled type 1 diabetics | Yes | 179 | 43.7 |
| | No | 231 | 56.3 |
| Dizziness | Yes | 227 | 55.4 |
| | No | 183 | 44.6 |
| Over all knowledge | Adequate | 190 | 46.3 |
| | Inadequate | 220 | 53.7 |

**Table 5. Attitude towards the antenatal exercise of respondents among pregnant women in Arba Minch town, southern, Ethiopia, 2020 (n = 410).**

| Attitudes regarding antenatal exercise | Strongly agree | | Agree | | Uncertain | | Disagree | | Strongly disagree | |
|---|---|---|---|---|---|---|---|---|---|---|
| | No. | % | No. | % | No. | % | No. | % | No. | % |
| Do you feel exercise during pregnancy is necessary? | 73 | 17.8 | 214 | 52.2 | 39 | 9.5 | 79 | 19.3 | 5 | 1.2 |
| Do you feel exercise during pregnancy is risky to the fetus? | 4 | 1 | 94 | 22.9 | 104 | 25.4 | 177 | 43.6 | 31 | 7.6 |
| Do you feel antenatal exercise suit with our culture? | 21 | 5.1 | 176 | 42.9 | 73 | 17.8 | 118 | 28.8 | 22 | 5.4 |
| Do you feel pregnant women should perform exercise under the guidance of health care professionals? | 91 | 22.2 | 172 | 42.0 | 81 | 19.8 | 60 | 14.6 | 6 | 1.5 |
| Do you feel ANEx can reduce pregnancy related complication? | 57 | 13.9 | 188 | 45.9 | 87 | 21.2 | 75 | 18.3 | 3 | 0.7 |
| Do you feel ANEx helps in post-delivery recovery? | 24 | 5.9 | 167 | 40.7 | 109 | 26.6 | 100 | 24.4 | 10 | 2.4 |
| Do you feel the exercising helps you get back to your shape? | 27 | 6.6 | 165 | 40.2 | 101 | 24.6 | 107 | 26.1 | 10 | 2.4 |
| Do you feel exercise regimen should vary from one to others? | 17 | 4.1 | 177 | 43.2 | 102 | 24.9 | 99 | 24.1 | 15 | 3.7 |
| Over all attitude | Unfavourable | | | | Number | | Percent (%) | | | |
| | | | | | 220 | | 54% | | | |
| | Favourable | | | | 190 | | 46% | | | |

ANEx-antenatal exercise, No.-number.

antenatal exercise during pregnancy is risky to the foetus. The median score value of pregnant women's attitude about antenatal exercise was 27. Among all participants only 46% 95%CI (42%-51%) pregnant women scored above mean value and had a favourable attitude towards antenatal exercise (Table 5).

## The practice of antenatal exercise among pregnant women

Nearly half 206(50.2%) of the respondents participated in physical exercise before becoming pregnant. Among total respondents, 135 (32.9%) respondents adequately practiced antenatal exercises while the majorities 275(67.1%) of the women practiced inadequately. Walking was the commonest type of exercise reported by 231 (89.9%) of the respondents. Most common reasons why pregnant women not practiced antenatal exercise in the current pregnancy were lack of information (32.9%) (Table 6, Fig 2).

## Factors associated with practices of antenatal exercise

Bivariable and multivariable binary logistic regression analyses were done to assess the association between the selected variables and the practice of antenatal exercise. During adjusted multivariable binary logistic regression analysis, six explanatory variables were significantly associated. From these, husband's primary school level, history of miscarriage, inadequate knowledge, and unfavourable attitude was positively associated with adequate antenatal exercise whereas, employment status of women and practice of regular exercises before current pregnancy were negatively associated (Table 7).

Those pregnant women who had husbands with primary schooling levels were 70% times less likely to practice adequate antenatal exercise than those whose husbands with college and above level educational status [AOR = 0.3, (95% CI: 0.1, 0.7)]. History of miscarriage is also negatively associated with practices of antenatal exercise. Those women who had a history of miscarriage were 70% less likely to practices antenatal exercise than those women without a history of miscarriage [A = 0.3, (95% CI: (0.1, 0.7)]. Having practices of regular exercise done before becoming pregnant was a risk to have antenatal exercise. Pregnant women with a practice of previous regular exercise were 1.9 times more likely to practice adequate antenatal exercise than those without a history of previous regular exercise [AOR = 1.9, (95% CI:1.1, 3.2)].

**Table 6. Practice of antenatal exercise among study participants in Arba Minch town, Southern, Ethiopia, 2020 (n = 410).**

| Variable | Response | Number | Percent (%) |
|---|---|---|---|
| Have you ever done regular exercise before becoming pregnant? | Yes | 206 | 50.2 |
| | No | 204 | 49.8 |
| Do you practice any antenatal exercise in this current pregnancy? | Yes | 257 | 62.7% |
| | No | 153 | 37.3% |
| If yes what type of exercise you exercising now? | Walking | 231 | 89.8% |
| | Aerobics | 2 | 0.7% |
| | Relaxation &breathing | 102 | 39.7% |
| | Pelvic floor exercise | 12 | 4.6% |
| | Back exercise | 76 | 29.6% |
| | Ankle and toe raising | 100 | 38.9% |
| | Stationary cycling | 1 | 0.4% |
| How many times per week did you exercise? | Less than two times/week | 122 | 47.8% |
| | Three and above/week | 135 | 52.2% |
| Minutes exercised per session | < 20 minutes | 122 | 47.8% |
| | ≥20minutes | 135 | 52.2% |
| Overall antenatal exercise practice | Inadequate practice | 275 | 67.1% |
| | Adequate practice | 135 | 32.9% |

Occupation of pregnant women had also an association with practices of antenatal exercise. Those employed pregnant women were 4.8 times more likely to practice adequate antenatal exercise than those who were unemployed [AOR = 4.8, (95% CI: 1.8, 13.0)]. Having inadequate knowledge and an unfavourable attitude to antenatal exercise also affects negatively the practices of antenatal exercise. Those pregnant women who had inadequate knowledge of antenatal exercise were 80% less likely to practice antenatal exercise than those who knew [AOR = 0.2,

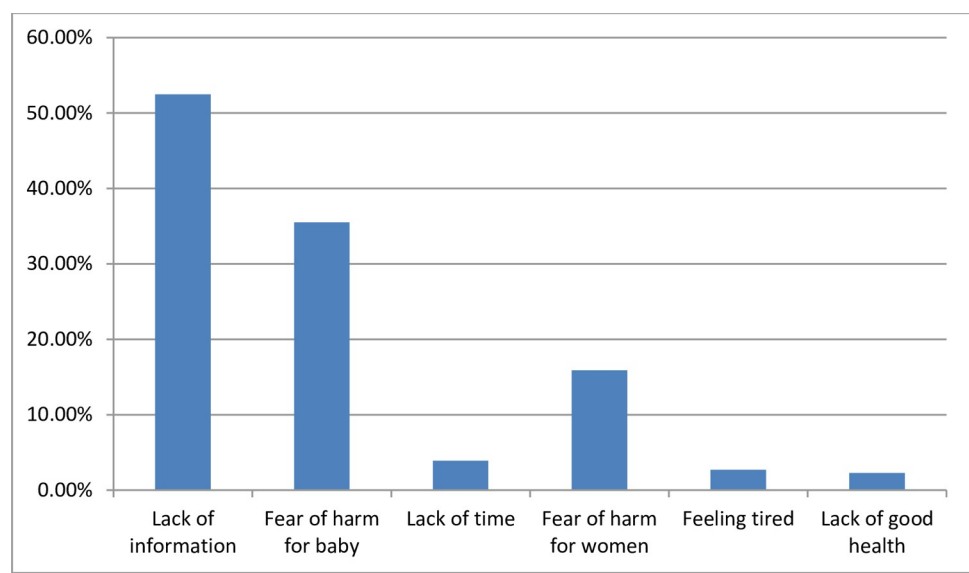

**Fig 2. Reasons for not practicing antenatal exercise in the current pregnancy among participants in Arba Minch town, Southern Ethiopia, 2020.**

**Table 7. Bi-variable and multi-variable analysis of factors associated with the practices of antenatal exercise in Arba Minch town, Southern Ethiopia, 2020 (N = 410).**

| Variables | Categories | Adequate antenatal exercise | | COR (95%CI) | AOR (95%CI) | P values |
|---|---|---|---|---|---|---|
| | | Yes | No | | | |
| Husband's educational status | Unable to read and write | 6 | 7 | 2.0 (0.6, 6.3) | 1.5 (0.1, 1.9) | |
| | Able to read and write | 15 | 37 | 0.9 (0.5, 1.9) | 0.7 (0.3, 1.9) | |
| | Primary school | **32** | **75** | **0.9 (0.6, 1 0.8)** | **0.3 (0.1, 0.7)** | **0.014** |
| | High school | 45 | 70 | 1.5 (0.9, 2.6) | 1.5 (0.4, 1.7) | |
| | College & above | 37 | 86 | 1 | 1 | |
| History of miscarriage | Yes | **6** | **33** | **0.3 (0.1, 0.8)** | **0.3 (0.1, 0.7)** | **0.010** |
| | No | 129 | 242 | 1 | 1 | |
| Regular exercise done before becoming pregnant. | Yes | **83** | **123** | **1.9 (1.3, 3.0)** | **1.9 (1.1, 3.2)** | **0.023** |
| | No | 52 | 152 | 1 | | |
| Knowledge | Inadequate | **43** | **177** | **0.3 (0.2, 0.4)** | **0.2 (0.1, 0.3)** | **0.001** |
| | Adequate | 92 | 98 | 1 | 1 | |
| Attitude | Unfavorable | **30** | **189** | **0.13 (0.08, 0.21)** | **0.3 (0.2, 0.5)** | **0.003** |
| | Favorable | 105 | 86 | 1 | 1 | |
| Age (years) | < 25 | 53 | 124 | 0.8 (0.5, 1.2) | 1.2 (0.7, 2.3) | |
| | > 25 | 82 | 151 | 1 | 1 | |
| Occupation | Employed | **64** | **11** | **3.4(1.7, 6.7)** | **4.8 (1.8, 13.1)** | **0.002** |
| | Non-employed | 211 | 124 | 1 | 1 | |
| Gravidity | One time | 34 | 104 | 1.4 (0.7, 3.2) | 1.8 (0.1, 8.3) | |
| | 2–3 time | 91 | 127 | 3.2 (1.5, 6.6) | 1.0 (0.2,1.3) | |
| | More than 3 | 10 | 44 | 1 | 1 | |

1 = reference group, COR = Crude odds ratio, AOR = Adjusted odds ratios.

(95% CI: 0.1, 0.3)]. And also women with unfavourable attitudes were 70% less likely to practice antenatal exercise than those who had favourable attitudes towards antenatal exercise [AOR = 0.3 (95% CI 0.2, 0.5)] (Table 7).

# Discussion

This study focused on assessing the adequate practice of antenatal exercise and its associated factors among pregnant women. The overall adequate practice of antenatal exercise during the current pregnancy in this study setting was 32.9% (95% CI: 28%-37%). This finding is in line with the study done, in Gondar town Northern Ethiopia, Pakistan, and Ireland [15,16,21]. But, lower than the result of the studies conducted in Nigeria and Zambia [11,15,22]. This difference might be due to lack of experience of physical exercise before pregnancy, low level of knowledge of antenatal exercise, lack of motivation, lack of awareness, and health care provider counselling in this study setting.

The majority of the women practiced walking, relaxation/breathing, and ankle and toe exercise as predominant antenatal exercise types in pregnancy. Walking was the most common type of exercise mentioned by respondents. This finding is in agreement with a study outcome from Seri Lanka and Pakistan [14,15]. This may be attributed to the fact that the walking activity is easy to carry out and there are no costs or equipment involved when performing the activity. On another hand, the study conducted in Zambia found that aerobics is the predominantly prescribed antenatal exercise [23].

Stationary cycling was the least practiced type of antenatal exercise in this study. This finding coincides with previous studies done in Nigeria [20]. Stationary Cycling exercises are not common in Arba Minch might be due to the lack of fitness programs in the town, and maybe unaffordable.

Regarding the frequency and duration of antenatal exercise among those who practiced, the majority of the women in the present study exercised for less than three days weekly and less than 30 minutes duration of exercise per session. This is in agreement with the study conducted in Nigeria and Zambia [20,23]. However, some of the women exercised in line with standard recommendations of ≥ 30 minutes daily. This is below the standard recommended exercise level of ≥ 30 minutes on most days of the week [4].

The majority of the women (34.9%) were guided by themselves (self-prescriber) to practice the exercise and other people while the role of health professional was very low. Majority of the exercisers unfulfilled the standards of ACOG. To get the benefits of antenatal exercise it might be important to have the antenatal exercise guideline which will help health professionals and health extension workers to guide women to strengthen ongoing maternal health promotion and education interventions. The most principal reasons given by the women for not engaging in an antenatal exercise in current pregnancy were lack of information and recommendation from health care providers and fear of foetal and women risks.

The result of the multivariable binary logistic regression model revealed that having the habit of regular exercise before becoming pregnant, and women having occupation status of employed showed significant positive associations. Whereas, women having husbands with primary education level, women having a history of miscarriage, having inadequate knowledge and unfavourable attitude towards antenatal exercise showed significant negative associations.

This study showed that the practice of antenatal exercise had a significant association with exercise performance before pregnancy. The practice of antenatal exercise was higher among those women who did physical exercise before becoming pregnant when compared with those women who did not. This finding is in line with a study that reported from, Brazil, Gondar town Northern Ethiopia [13,16]. This may be due to women who had practiced antenatal exercise cannot get challenges in adherence to the advice of health professionals about antenatal exercise. It was found that the practice of antenatal exercise had a significant association with the occupation status of being employed. The antenatal exercise was highly practiced among those women who were employed when compared to the unemployed. This study finding is consistent with the research findings conducted in Zambia [23]. This might be those employed women can easily afford different information sources like TV, radio, smartphone which can help them to have the knowledge and a favourable attitude towards antenatal exercise and its health effects.

The education status of husbands had a statistically significant association with the practice of antenatal exercise. Those women who had husbands with a primary school level were 70% times less likely to practice an antenatal exercise as compared to those women whose husbands had education level of college and above. This might be because highly educated husbands can have a good awareness of the importance of antenatal exercise for women and their foetuses than less educated and uneducated husbands. Therefore, these educated husbands can help their wives to engage in antenatal exercise. In this society, there are unique cultural norms and values such as the respect of husband's advice by their wives which affects the decisions of women.

Those women who had ever history of miscarriage were less likely to practice antenatal exercise compared to those women who did not have. This might be due to safety concerns since they once had a miscarriage before; more precautions may be taken due to fear of risk (harm) to their foetus. Those women who had inadequate knowledge of antenatal exercise

were less likely to practice antenatal exercise when compared to those women with adequate knowledge. This finding was in line with the report of a study done in Gondar town, Northern Ethiopia [16]. This might be due to, the finding of this study revealed that, from total respondents, 53.7% had inadequate knowledge about antenatal exercise. These individuals may not know the benefits of ANEx for themselves and their foetuses. Likewise, women with unfavourable attitudes were less likely to practices antenatal exercise when compared to women having favourable attitudes to practices of antenatal exercise. This finding was in agreement with the report of a study done in Gondar town, Northern Ethiopia [16]. This may be due to the feeling of women towards the benefits and contraindications of ANEx. The finding of this study showed that about 54% of participants had unfavourable attitudes which means they had negative attitudes towards the benefits of ANEx so that they cannot adhere to practices of adequate antenatal exercise.

## Strengths and limitations of the study

This study was community-based which can make the findings more generalizable to the pregnant women in the town. It is difficult to establish a temporal relationship as the study design was cross-sectional. Practice evaluation tools need to be observation checklists, but interview questions used to this may reduce the quality and adequacy of participants' antenatal exercise judgment.

Despite these limitations, the findings from this study will contribute to the understanding of the factors associated with adequate practices of antenatal exercise in the study area.

## Conclusion and recommendations

In this study, the adequate practice of antenatal exercises was found to be low. Husband's primary school level, history of miscarriage, inadequate knowledge, and unfavourable attitude was negatively associated with adequate antenatal exercise whereas, the employment status of women and practice of regular exercise before current pregnancy was positively associated with adequate antenatal exercise. Appropriate measures should be taken to improve the husband's educational level, mother's occupation, knowledge, and attitudes towards antenatal exercise. Special consideration should be given to those with a history of miscarriage and women should be encouraged to have practices of regular exercise before pregnancy. And also further study should be conducted by triangulating with a qualitative study.

## Supporting information

**S1 File. Questionnaire for the study of factors associated with antenatal exercise in Arba Minch town, Southern Ethiopia.**
(DOCX)

**S2 File. SPSS data set for the study of factors associated with antenatal exercise in Arba Minch town, Southern Ethiopia.**
(SAV)

## Acknowledgments

The authors acknowledge the Arba Minch town Kebele leaders for their cooperation and pregnant women for providing the data. We also would like to acknowledge data collectors and supervisors for accomplishing their tasks.

## Author Contributions

**Conceptualization:** Maechel Maile Beyene, Mulugeta Shegaze Shimbre, Gebresilasea Gendisha Ukke, Mathewos Alemu Gebremichael, Mekdes Kondale Gurara.

**Data curation:** Maechel Maile Beyene, Mulugeta Shegaze Shimbre, Gebresilasea Gendisha Ukke, Mathewos Alemu Gebremichael, Mekdes Kondale Gurara.

**Formal analysis:** Maechel Maile Beyene, Mulugeta Shegaze Shimbre, Gebresilasea Gendisha Ukke, Mathewos Alemu Gebremichael, Mekdes Kondale Gurara.

**Funding acquisition:** Maechel Maile Beyene.

**Methodology:** Maechel Maile Beyene, Mulugeta Shegaze Shimbre, Gebresilasea Gendisha Ukke, Mathewos Alemu Gebremichael, Mekdes Kondale Gurara.

**Resources:** Maechel Maile Beyene.

**Supervision:** Maechel Maile Beyene, Mulugeta Shegaze Shimbre, Gebresilasea Gendisha Ukke, Mathewos Alemu Gebremichael.

**Writing – original draft:** Maechel Maile Beyene.

**Writing – review & editing:** Maechel Maile Beyene, Mulugeta Shegaze Shimbre, Gebresilasea Gendisha Ukke, Mathewos Alemu Gebremichael, Mekdes Kondale Gurara.

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
