## [Decision Letter · Decision Letter 0]

14 Aug 2021

PONE-D-21-11918

Practice of antenatal exercise and its associated factors among pregnant women in Arba Minch town, Southern Ethiopia: A community based cross-sectional study

PLOS ONE

Dear Dr. Beyene,

Thank you for submitting your manuscript to PLOS ONE. After careful consideration, we feel that it has merit but does not fully meet PLOS ONE’s publication criteria as it currently stands. Therefore, we invite you to submit a revised version of the manuscript that addresses the points raised during the review process.

Reviewer 2 has indicated a number of points regarding your methods and the presentation of your manuscript that should be addressed. Please attend carefully to each of their points when preparing your revisions.

We look forward to receiving your revised manuscript.

Kind regards,

Jamie Males

Staff Editor

PLOS ONE

Journal Requirements:

4. Please include a caption for figure 1.

5. Please include your tables as part of your main manuscript and remove the individual files. Please note that supplementary tables (should remain/ be uploaded) as separate "supporting information" files

Reviewers' comments:

Reviewer's Responses to Questions

**Comments to the Author**

1. Is the manuscript technically sound, and do the data support the conclusions?

Reviewer #1: Yes

Reviewer #2: Yes

2. Has the statistical analysis been performed appropriately and rigorously? 

Reviewer #1: Yes

Reviewer #2: Yes

3. Have the authors made all data underlying the findings in their manuscript fully available?

Reviewer #1: Yes

Reviewer #2: Yes

4. Is the manuscript presented in an intelligible fashion and written in standard English?

Reviewer #1: Yes

Reviewer #2: Yes

5. Review Comments to the Author

Reviewer #1: The authors have addressed all the comments. As indicated in my earlier comments, given the paucity of information on prenatal physical activity in Africa, this study would contribute to the body of knowledge on prenatal physical activity and exercise in Africa.

Reviewer #2: Thank you for the opportunity to review this manuscript. The approach is not new, however, it brings data from a population that has been little studied in relation to the practice of PE and with that the material becomes more interesting.

Title – very long - Practice of antenatal exercise and its among pregnant women in Arba Minch town, Southern Ethiopia: A community based cross-sectional study

Suggestion - Factors associated with antenatal exercise in Arba Minch town, Southern Ethiopia: A community based cross-sectional study

Abstract -

I suggest that the results with positive or negative association according to the PE practice be presented separately, both in the results and in the conclusion

Ethics – “informed consent was taken” - Written or verbal? If verbal, how was this obtained and recorded?

Introduction

– please update your data

Reference 3 - according to article - Review of Recent Physical Activity Guidelines During Pregnancy to Facilitate Advice by Health Care Providers. Evenson KR, Mottola MF, Artal R.Obstet Gynecol Surv. 2019 Aug;74(8):481-489. doi: 10.1097/OGX.0000000000000693.

Reference 7 - according to article - Physical Activity Patterns and Factors Related to Exercise during Pregnancy: A Cross Sectional Study. Nascimento SL, Surita FG, Godoy AC, Kasawara KT, Morais SS.PLoS One. 2015 Jun 17;10(6):e0128953. doi: 10.1371/journal.pone.0128953. eCollection 2015.

Method:

Sample size – “adequate practice of 95 antenatal exercise to be 50%” - Very high 50%, what is the reference used?

I also suggest that you include a figure that explains the representativeness of the sample in different districts, sub-districts, this was not made explicit and can be very well explained with a figure or even a map.

Results

– Will the tables not be shown in the results? They are all in the supporting information. I believe it is fundamental to see the numbers of the data presented and the tables must appear in the text, especially 5, 6 and 7 with the analyses.

And it needs to be very clear which associations are positive and which are negative (this in the results, discussion and conclusion)

Discussion

It also needs to contextualize more in the discussion the importance of marriage, education of the woman and the husband in the society where the study was carried out.

And in the limitations, further reinforce the question of the representativeness of the sample, that is, how much it represents the city, region or country where the study was carried out.

6. PLOS authors have the option to publish the peer review history of their article (what does this mean?). If published, this will include your full peer review and any attached files.

Reviewer #1: No

Reviewer #2: No

---

## [Author Response · Author response to Decision Letter 0]

28 Sep 2021

Thank you for your valuable comments to improve this article

---

## [Decision Letter · Decision Letter 1]

18 Nov 2021

Factors associated with antenatal exercise in Arba Minch town, Southern Ethiopia: a community-based cross-sectional study

PONE-D-21-11918R1

Dear Dr. Beyene,

We’re pleased to inform you that your manuscript has been judged scientifically suitable for publication and will be formally accepted for publication once it meets all outstanding technical requirements. Please also note one of the reviewer identified an important text change in the conclusion section (see comments appended below) that you will need to address during our final technical checks  

Kind regards,

Dario Ummarino, Ph.D.

Senior Editor

PLOS ONE

Additional Editor Comments (optional):

Reviewers' comments:

Reviewer's Responses to Questions

**Comments to the Author**

1. If the authors have adequately addressed your comments raised in a previous round of review and you feel that this manuscript is now acceptable for publication, you may indicate that here to bypass the “Comments to the Author” section, enter your conflict of interest statement in the “Confidential to Editor” section, and submit your "Accept" recommendation.

Reviewer #1: (No Response)

Reviewer #2: All comments have been addressed

2. Is the manuscript technically sound, and do the data support the conclusions?

Reviewer #1: (No Response)

Reviewer #2: Yes

3. Has the statistical analysis been performed appropriately and rigorously? 

Reviewer #1: (No Response)

Reviewer #2: Yes

4. Have the authors made all data underlying the findings in their manuscript fully available?

Reviewer #1: (No Response)

Reviewer #2: Yes

5. Is the manuscript presented in an intelligible fashion and written in standard English?

Reviewer #1: (No Response)

Reviewer #2: Yes

6. Review Comments to the Author

Reviewer #1: Thank you for the quick response. The authors have attended to the comments to my satisfaction. The manuscript can be accepted.

Reviewer #2: The presentation of results at the conclusion of the article is reversed - this needs to be corrected - see-

"Husband’s primary school level, history of miscarriage, inadequate knowledge, and unfavourable attitude was positively associated with adequate antenatal exercise whereas, the employment status of women and practice of regular exercise before current pregnancy was negatively associated with adequate antenatal exercise."

7. PLOS authors have the option to publish the peer review history of their article (what does this mean?). If published, this will include your full peer review and any attached files.

Reviewer #1: **Yes: **Prof Daniel Ter Goon

Reviewer #2: No

---

## [Editor Report · Acceptance letter]

13 Dec 2021

PONE-D-21-11918R1 

Factors associated with antenatal exercise in Arba Minch town, Southern Ethiopia: a community-based cross-sectional study 

Dear Dr. Beyene:

I'm pleased to inform you that your manuscript has been deemed suitable for publication in PLOS ONE. Congratulations! Your manuscript is now with our production department. 

Kind regards, 

on behalf of

Dr. Dario Ummarino 

Staff Editor

PLOS ONE